# Gender-Specific Impacts of the COVID-19 Pandemic on Orthopedic and Traumatology Care: An Analysis of Hospital Admissions and Length of Stay

**DOI:** 10.3390/healthcare12202031

**Published:** 2024-10-12

**Authors:** Karoly Bancsik, Lucia Georgeta Daina, László Lorenzovici, György Rossu, Raluca Bancsik, Timea Claudia Ghitea, Codrin Dan Nicolae Ilea, Mădălina Diana Daina

**Affiliations:** 1Faculty of Medicine and Pharmacy, Doctoral School of Biomedical Sciences, University of Oradea, 1 December Sq., 410081 Oradea, Romania; karoly.bancsik@didactic.uoradea.ro (K.B.); rossugy@yahoo.com (G.R.); codrin.ilea@csud.uoradea.ro (C.D.N.I.); 2Department of Psycho-Neurosciences and Recovery, Faculty of Medicine and Pharmacy, University of Oradea, 1 December Sq., 410081 Oradea, Romania; lucidaina@gmail.com; 3Department of Doctoral Studies, George Emil Palade University of Medicine, Pharmacy, Science, and Technology of Târgu Mureș, 540142 Tirgu Mures, Romania; lorenzovici@syreon.ro; 4Clinical Emergency Hospital “Avram Iancu”, 410027 Oradea, Romania; ralu_iles@yahoo.com; 5Pharmacy Department, Faculty of Medicine and Pharmacy, University of Oradea, 1 December Sq., 410081 Oradea, Romania; 6Faculty of Medicine and Pharmacy, University of Oradea, 1 December Sq., 410081 Oradea, Romania; daina.madalinadiana@student.uoradea.ro

**Keywords:** COVID-19, pandemic, hospitalization, gender-specific, orthopedics, traumatology

## Abstract

Background: Understanding gender disparities in healthcare outcomes is crucial, especially during crises like the COVID-19 pandemic. The absence of gender-specific data on hospital admissions and lengths of stay for orthopedic and traumatology patients limits the precision of current analyses, making conclusions speculative. Objective: This study aims to highlight the potential insights that could be gained from gender-disaggregated data, illustrating how a more precise gender-based analysis could reveal healthcare disparities in orthopedic and trauma care during the pandemic. Materials and Methods: A robust analysis would require gender-disaggregated data, including variables such as admission rates, lengths of stay, injury types, and access to care, along with factors like age and socioeconomic status. In the absence of such data, a hypothetical framework was developed based on known healthcare disparities, using general trends to project possible gender-specific impacts. Results: Without gender-disaggregated data, it remains challenging to identify specific gender differences in outcomes accurately. Hypothetical scenarios suggest that disparities in admission rates, hospital stay durations, and access to care could be influenced by factors such as increased caregiving responsibilities for women or differential access to healthcare based on socioeconomic status. Conclusions: To accurately assess gender disparities in orthopedic and trauma care during the pandemic, future studies must prioritize the collection and use of gender-disaggregated data. This approach is essential for drawing reliable conclusions and developing targeted interventions to address healthcare inequities effectively.

## 1. Introduction

Prior to the COVID-19 pandemic, significant research highlighted gender disparities in healthcare utilization, access, and outcomes. Studies have shown that men and women experience healthcare differently due to a combination of biological, social, and behavioral factors. Men are generally less likely to seek preventive care and more likely to delay medical consultation, leading to higher rates of acute hospital admissions for conditions that could have been managed earlier [1,2]. This trend is particularly evident in orthopedic and trauma care, where men have historically been overrepresented in trauma cases due to higher rates of occupational and recreational injuries [3].

In contrast, women are more likely to suffer from chronic musculoskeletal conditions such as osteoporosis and arthritis, often leading to higher elective surgery rates, including joint replacements [3,4]. Despite these differences, women often face barriers to accessing specialized care and may experience longer wait times for surgical interventions [5]. The pandemic, therefore, presents a unique opportunity to explore how these pre-existing gender disparities have evolved under the strains imposed by COVID-19.

Previous research has also demonstrated that healthcare crises tend to exacerbate existing inequalities, disproportionately affecting vulnerable populations, including women and men with differing healthcare needs [6]. Given this context, our study aims to build on the existing literature by examining how the COVID-19 pandemic has impacted gender differences in hospitalizations, lengths of stay, and access to orthopedic and traumatology care. By comparing pre-pandemic and pandemic data, we aim to provide a comprehensive analysis of how these disparities have shifted during this unprecedented period.

Gender disparities in healthcare have long been documented, with research showing that men and women experience different health outcomes due to biological, social, and behavioral factors [7,8]. However, the lack of gender-disaggregated data in many healthcare studies poses a significant limitation, obscuring the true extent of these disparities and hindering targeted interventions [9]. This issue becomes even more critical during health crises like the COVID-19 pandemic, where resource allocation, risk mitigation, and patient outcomes can vary significantly between genders.

Without detailed, gender-specific data, healthcare systems risk implementing one-size-fits-all approaches that fail to address the unique needs of male and female patients. For example, the pandemic has affected men and women differently, not only in terms of infection rates and clinical outcomes but also in access to healthcare services [6]. Men, who are often less likely to seek preventive care, may experience delays in receiving necessary treatment for non-COVID conditions, as shown by the significant reduction in orthopedic and trauma admissions among male patients during the pandemic. Conversely, women may face increased barriers to accessing elective procedures, leading to a worsening of chronic conditions [10].

The absence of gender-disaggregated data thus limits our ability to identify these nuanced differences and to develop effective, equitable healthcare policies. By analyzing hospital admissions and outcomes in orthopedics and traumatology with a gender lens, this study not only addresses this critical gap but also highlights the importance of incorporating gender-specific analyses into future healthcare research. Doing so is essential for ensuring that healthcare systems are resilient and responsive to the needs of all patient populations, particularly in times of crisis. Despite providing valuable insights, the lack of comprehensive gender-disaggregated data limits the depth of our analysis. More detailed information on patient comorbidities, socioeconomic status, and healthcare-seeking behavior would have enabled a fuller understanding of the gender disparities observed. However, our findings still reveal significant gender-specific trends in hospitalizations and lengths of stay, demonstrating that even limited data can uncover important patterns. This highlights the need for more robust data collection practices that include gender as a key variable. Future research should focus on gathering such data to better address gender-specific healthcare needs, particularly during crises like the COVID-19 pandemic. Hospitalizations for chronic conditions, especially among male patients, and the average length of hospital stays have become critical indicators of the pandemic’s effects on healthcare systems, revealing significant gender disparities [11,12].

The pandemic’s impact on hospital admissions within orthopedics and traumatology departments has been both substantial and complex. Shifts in patient volumes and acuity levels have underscored the challenges of healthcare delivery during this global crisis. Men, who comprised the majority of the patient population, experienced notable variations in admission rates and lengths of stay, reflecting the broader difficulties faced by the healthcare system [13,14,15,16].

Our study contributes to the theoretical understanding of how global health crises, such as the COVID-19 pandemic, can exacerbate existing gender disparities in healthcare access and outcomes. By highlighting these gender-specific trends in orthopedic and traumatology departments, our findings emphasize the importance of incorporating a gender perspective into health system research and crisis management. Practically, this study suggests that healthcare policies and emergency response plans should be tailored to address the distinct needs of male and female patients, ensuring equitable access to care during and beyond pandemic periods. This approach could improve healthcare delivery and outcomes, particularly for vulnerable populations.

This study presents a retrospective analysis of orthopedic and trauma cases admitted during the COVID-19 pandemic, covering the pre-pandemic, pandemic, and post-pandemic periods over an eight-year timeframe (2015–2022) [17,18]. Research has shown that gender differences significantly influence healthcare utilization, access, and outcomes. For instance, men are more likely to delay seeking medical care and have higher rates of severe health conditions upon admission, which may contribute to longer hospital stays [1,2]. Women, on the other hand, often face barriers to accessing specialized care and may experience longer wait times for elective procedures [8]. These disparities are further exacerbated during health crises, as healthcare systems prioritize resources and may inadvertently widen the gap between genders [6]. By incorporating a gender perspective, this study aims to provide a nuanced understanding of how the pandemic affected male and female patients differently, emphasizing the importance of considering gender as a key variable in healthcare research. The objective is to analyze the impact of the COVID-19 pandemic on hospital management indicators in orthopedics and traumatology departments, with a specific focus on identifying gender differences in hospitalizations, lengths of stay, and access to care among male and female patients. The research also examines how the strategic restructuring and reorganization of hospitals in Bihor County, implemented in response to the pandemic, influenced these indicators. The analysis includes the number of hospitalizations in orthopedics and traumatology departments at the County Clinical Emergency Hospital Oradea, broken down by type of hospitalization, with a focus on gender differences. Additionally, the study assesses the average length of hospital stays and bed availability in these departments, identifying gender-specific trends through annual reporting.

## 2. Materials and Methods

### 2.1. Study Design

This research is a retrospective observational study conducted at the County Clinical Emergency Hospital Oradea. Data collection for this study was conducted in strict accordance with ethical standards to ensure the protection of patient confidentiality and the integrity of the research. Authorization to access hospital records was granted by the Institutional Review Board (IRB) of the County Clinical Emergency Hospital Oradea (CCEHO). All patient data were anonymized prior to analysis to safeguard personal information. The dataset included all admissions to the orthopedics and traumatology departments from 2015 to 2022, with no exclusion criteria applied, thereby maintaining the comprehensiveness of the analysis. This adherence to ethical guidelines was in compliance with national and international standards for research involving human subjects, ensuring that the study was conducted with the highest level of ethical responsibility [19,20,21].

The study was carried out at CCEHO in northwestern Romania, a public healthcare institution that serves approximately 200,000 residents of Oradea and provides emergency care to a broader population of around 600,000 people in the surrounding region. This research is part of a broader doctoral project aimed at evaluating medical services within the orthopedics and traumatology departments at CCEHO, with a particular focus on gender disparities [22]. Previous analyses indicated a significant reduction in hospital admissions beginning in April 2020, coinciding with the onset of the COVID-19 pandemic. This reduction particularly affected male patients, who accounted for 56% of admissions [23]. The current study extends this analysis to cover the pre-pandemic, pandemic, and post-pandemic periods (2015–2022), with an emphasis on gender-specific outcomes.

### 2.2. Data Collection and Participants

After obtaining authorization to access hospital records, data were retrieved from the CCEHO Statistical Department. The dataset included all admissions to the orthopedics and traumatology departments from 2015 to 2022, covering a total of 19,950 patients, 56% of whom were male. No exclusion criteria were applied to the dataset. To provide a broader context, data on confirmed COVID-19 cases in Bihor County from 2020 to 2022, totaling 92,159 cases, were also incorporated. These data were publicly available from the Romanian Ministry of Health and the Covid19.stirioficiale.ro project by the Code for Romania Task Force [24].

Romania’s healthcare system is a predominantly public system funded through the National Health Insurance Fund, which provides coverage to the majority of the population [25]. However, access to hospital services can be uneven, particularly in rural areas, and there are disparities in the availability of specialized care. The healthcare infrastructure, including the number of hospital beds and healthcare professionals, is often concentrated in urban centers, which can lead to differences in access to care between urban and rural populations [25]. During the COVID-19 pandemic, these pre-existing challenges were exacerbated by the strain on healthcare resources, which may have contributed to the observed declines in hospital admissions and changes in care patterns. Understanding these systemic characteristics is essential for interpreting the gender-specific outcomes of this study, as they may reflect not only the direct impact of the pandemic but also the broader limitations and inequities within the healthcare system.

### 2.3. Statistical Analysis

Linear regression analyses were employed to assess the impact of the COVID-19 pandemic on both the number of hospitalizations in the orthopedics and traumatology departments and the average length of stay [26]. Results were considered statistically significant if the *p*-value was less than 0.05. Data processing and statistical analysis were conducted using Microsoft Word and Excel software (Profesional Plus 2021, Version 2409) tools. Potential selection bias was addressed by including all patients admitted to the orthopedics and traumatology departments between 2015 and 2022 without applying exclusion criteria. This approach aimed to capture the full spectrum of admissions during the study period. Missing data were minimal (<5%) and were handled through listwise deletion, as sensitivity analyses indicated no significant impact on the results. To account for confounding factors, such as age and underlying health conditions, stratified analyses were performed, and results were cross-validated using multivariate regression models.

## 3. Results

### 3.1. Hospital Admissions

Between 2015 and 2019, the orthopedics and traumatology departments experienced a steady number of inpatient admissions, with yearly figures fluctuating between 2500 and 2800 cases. However, as illustrated in Table 1, the year 2020 marked a significant downturn in hospital admissions, plummeting by 36.5% due to the onset of the COVID-19 pandemic. This decline disproportionately impacted male patients, who comprised 56% of the total admissions. The following year, 2021, saw a gradual recovery in admission numbers, showing a 9.5% increase. By 2022, the number of hospitalizations had fully rebounded to pre-pandemic levels, indicating a return to normalcy in patient activity.

The number of patients admitted for continuous stays also experienced a significant drop of 33.9% in 2020, followed by a slow recovery in 2021 and 2022, with men continuing to represent the majority of patients. Admissions for one-day stays saw an even more drastic decline, dropping by 76.7% in 2020. In 2021, one-day admissions were nearly nonexistent, with only a slight resumption in 2022.

The statistical data highlight a pronounced link between the sharp decline in admissions to the orthopedics and traumatology departments and the emergence of the COVID-19 pandemic in 2020 (*p* < 0.001). These results reflect the considerable disruptions caused by the pandemic on hospital operations, with a notable impact on male patients. The shifting healthcare needs and restrictions during the pandemic period significantly altered the usual patterns of hospital admissions, illustrating the far-reaching effects of the global health crisis.

Table 2 summarizes the data across the years, highlighting the percentage changes and trends for both continuous stay and one-day admissions, reflecting a sharp decline in 2020 due to the pandemic, followed by a gradual recovery in subsequent years.

Between 2015 and 2019 (Table 3), hospital admissions to the orthopedics and traumatology departments remained relatively stable, averaging around 2500 cases annually, with a consistent male predominance of 56%. However, this stability was disrupted in 2020, with a significant 13% decline in admissions, dropping to 1698 cases. This decrease strongly correlates with the onset of the COVID-19 pandemic, which disproportionately affected male patients. Urgent case admissions saw a notable 29.6% reduction in 2020, primarily impacting male patients. Although a gradual recovery began in 2021, overall admissions still remained below pre-pandemic levels.

In contrast, chronic patient admissions plummeted by 51.8% in 2020 but showed a strong rebound in subsequent years, increasing by 30–45% annually for both male and female patients. Statistical analysis confirmed a strong correlation (*p* < 0.001) between the spike in COVID-19 cases in 2020 and the decline in both urgent and chronic admissions, particularly among male patients. This correlation weakened in 2021 and 2022 as admission patterns began to normalize.

One-day admissions in the orthopedics and traumatology departments showed a consistent trend from 2015 to 2019, largely serving chronic patients, with nearly 99.6% of these cases being non-urgent. However, during 2020 and 2021, the number of these admissions plummeted dramatically, falling from 163 in 2019 to just 38 in 2020 and further down to 4 in 2021. This sharp decline was directly linked to the surge in COVID-19 cases and associated healthcare restrictions. By 2022, the number of one-day admissions began to recover, rising to 59 cases, but the impact of the pandemic on these short-stay hospitalizations remained evident, as shown by the statistical correlation analysis (*p* < 0.001) in Figure 1.

### 3.2. Average Length of Hospitalization and Available Beds

The average duration of hospitalization for patients in the orthopedics and traumatology departments at CCEHB fluctuated over the course of the study, with notable disparities observed between male and female patients.

In 2020, the ALS experienced a substantial decrease of 16.6%, with this reduction being more pronounced among male patients. This downward trend continued in 2021 with an additional 4.7% decrease, again predominantly affecting males. Conversely, 2022 saw an 11.1% increase in ALS, indicating a resurgence in care needs for both male and female patients.

Bed availability within these departments also fluctuated during the study period. In 2020, there was a 12.7% reduction in available beds, which returned to pre-pandemic levels in 2021 and 2022. These changes in bed capacity impacted male and female patients differently across the respective years.

Statistical analysis showed a significant link between the number of COVID-19 cases in 2020 and 2021 and the average hospital stay in the orthopedics and traumatology departments (*p* < 0.001), particularly impacting male patients. Table 4 presents the annual ALS distribution in relation to COVID-19 cases in Bihor County.

Between 2020 and 2022 (Table 5), COVID-19 case numbers in Bihor County showed significant increases, with distinct gender differences; in 2020, there were 17,689 cases (8137 female and 9552 male); in 2021, cases surged to 33,374 (14,685 female and 18,689 male); and by 2022, the total reached 41,099 (16,851 female and 24,248 male).

Bed availability in the orthopedics and traumatology departments at CCEHB remained steady at 63 beds per year from 2015 to 2019. In 2020, it dropped by 12.7%, significantly correlated with the surge in COVID-19 cases (*p* < 0.001), particularly affecting male patients. By 2021 and 2022, bed availability returned to 63 beds, indicating successful adaptation to balance care for both COVID-19 and non-COVID patients despite the ongoing pandemic.

## 4. Discussion

The COVID-19 pandemic posed unprecedented challenges to healthcare systems globally, significantly disrupting services provided by orthopedics and traumatology departments [27]. This discussion focuses on the pandemic’s impact on hospital admissions and lengths of stay, highlighting gender-specific patterns [28]. By analyzing these trends, the aim is to shed light on the disruptions and adaptations within these departments during the pandemic [29].

One possible explanation for the higher impact on male admissions and extended hospital stays could be cultural and occupational factors. Men are often more involved in high-risk occupations and physical activities, which increases their likelihood of sustaining injuries requiring orthopedic and trauma care [27,30]. Additionally, cultural attitudes towards healthcare-seeking behavior may contribute, as men are generally less likely to seek timely medical attention, potentially leading to more severe conditions by the time they are admitted [1]. Socioeconomic factors, such as employment in physically demanding jobs and less access to health insurance, may also play a role, as these conditions may limit their ability to take leave for elective procedures, leading to extended hospital stays when care is finally sought [2]. These factors, combined with the disruption of healthcare services during the pandemic, likely contributed to the observed gender disparities.

During the pandemic, orthopedics and traumatology departments experienced a sharp decline in admissions for elective procedures, such as joint replacements and arthroscopic surgeries, as hospitals prioritized COVID-19 care [31]. This reduction was particularly evident among male patients, who are more commonly recipients of these procedures. Additionally, many patients, regardless of gender, postponed seeking treatment for musculoskeletal issues due to concerns about the virus [32].

While our findings indicate a significant decline in hospital admissions during the pandemic, it is important to consider that this trend may not be solely attributable to COVID-19 itself. Several factors, including public fear of contracting the virus, reduced access to healthcare due to lockdown measures, changes in healthcare policies prioritizing COVID-19 cases, and the reallocation of medical resources, could have influenced the observed changes in hospital care. As such, we interpret the association between the pandemic and the reduction in admissions with caution and acknowledge that further studies are needed to explore these potential contributing factors in more detail. This nuanced understanding is crucial for developing effective healthcare strategies in response to future crises.

Our findings highlight the need for integrating gender analysis into healthcare crisis response strategies, as the pandemic has exacerbated existing disparities in access and outcomes. Male patients were disproportionately affected by reduced access to elective procedures and increased trauma-related admissions [4,33,34]. To mitigate these issues in future crises, healthcare systems should focus on systematically collecting and reporting gender-disaggregated data, allocating resources equitably based on specific gender needs, and developing tailored health communication strategies that encourage timely healthcare access. Implementing these measures can ensure more equitable care and better outcomes for all patient populations during health emergencies [35,36,37,38].

Despite the decline in elective surgeries during the pandemic, the need for urgent and trauma care in orthopedics remained consistent, with variations based on gender. Traumatic injuries, such as fractures and dislocations, continued to require immediate medical attention. Lockdown measures and restrictions may have contributed to an increase in certain types of trauma, such as falls and sports-related injuries, with male patients often being more affected. This rise in trauma cases added to the workload in orthopedic departments, highlighting gender-specific trends [39].

Our findings on the decline in orthopedic and traumatology admissions during the COVID-19 pandemic are consistent with similar studies conducted in other countries. For example, studies from Italy and Spain reported significant reductions in elective orthopedic procedures and trauma cases attributed to lockdown measures and the reallocation of healthcare resources [40,41]. Additionally, a study from the United States found that orthopedic trauma admissions decreased while the severity of cases admitted increased, suggesting a delay in seeking care due to pandemic-related concerns [42].

Comparing these results with our study, we observe a similar trend of reduced admissions and increased lengths of stay, particularly among male patients. However, unlike some other countries, the healthcare system in Romania faced specific challenges related to resource availability and access to specialized care, which may have contributed to the observed gender disparities. These findings highlight the importance of considering local healthcare contexts when interpreting pandemic-related healthcare disruptions.

Future studies could explore how different healthcare systems adapted to the pandemic and whether these adaptations influenced the observed outcomes. Such comparisons could provide valuable insights for improving healthcare resilience in the face of future crises.

The pandemic also influenced the length of hospital stays for orthopedic and traumatology patients. Hospitals implemented strategies to shorten stays and expedite discharges to manage bed capacity and reduce the risk of COVID-19 transmission [43]. Elective surgery patients, regardless of gender, were often discharged earlier to free up beds. However, postponements of non-urgent procedures sometimes led to longer hospital stays for patients waiting for treatment, with male patients occasionally facing extended delays [44,45,46,47].

### Limitations of the Study

This analysis focuses on the orthopedics department of a single hospital that was designated as a COVID-19 support facility during the pandemic. To accommodate the surge in COVID-19 cases, this hospital reduced its orthopedics department bed capacity by eight beds. The data span from 2015 to 2022, providing a detailed view of hospitalization patterns based on service types and examining the relationship between these hospitalizations and COVID-19 case counts.

Several limitations should be considered when interpreting our results. Firstly, the lack of comprehensive gender-disaggregated data limits our ability to explore all underlying factors contributing to the observed disparities. Additionally, potential confounding variables, such as socioeconomic status and comorbidities, were not fully accounted for, which may influence the findings. The study is also limited to a single healthcare institution, which may affect the generalizability of the results to other settings or countries.

Future research should aim to address these limitations by collecting more detailed, gender-disaggregated data and incorporating a broader range of variables, such as socioeconomic factors and health-seeking behaviors. Comparative studies across different healthcare settings and countries would be valuable to understand how varying healthcare systems influence gender disparities during health crises. Moreover, longitudinal studies could help assess the long-term impacts of the pandemic on healthcare access and outcomes for different patient populations.

## 5. Conclusions

Future studies should aim to address the limitations identified in our research, particularly the lack of comprehensive gender-disaggregated data. By filling these gaps, more detailed investigations can provide deeper insights into the underlying factors contributing to gender disparities in healthcare. Our study serves as an initial step in highlighting the importance of integrating gender analysis into healthcare research and policy planning, especially in response to health crises like the COVID-19 pandemic. Further research is essential to develop effective strategies that ensure equitable healthcare for all populations. The strong correlation between the pandemic’s onset and changes in healthcare delivery, particularly among male patients, highlights the significant strain on resources and the sharp decline in elective procedures. Despite these challenges, the need for urgent and trauma care persisted, necessitating adaptive strategies that led to both shortened and extended hospital stays depending on circumstances. These findings emphasize the critical importance of maintaining adaptable healthcare strategies that can address both the immediate demands of a crisis and the unique needs of different patient populations, ensuring effective and equitable care.

## Figures and Tables

**Figure 1 healthcare-12-02031-f001:**
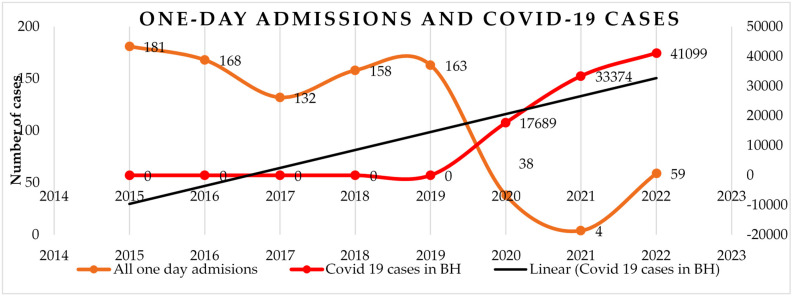
Distribution of one-day admissions according to urgency assessed annually in correlation with COVID-19 cases in Bihor County.

**Table 1 healthcare-12-02031-t001:** Demographic characteristics of the study cohort.

Characteristic	N (%)			
	Sex		Residence	
Year	Male	Female	Urban	Rural
2017	1479 (58%)	1071 (42%)	1228 (48%)	1322 (52%)
2018	1017 (41%)	1463 (59%)	1152 (46%)	1328 (54%)
2019	1378 (53%)	1222 (47%)	1276 (49%)	1324 (51%)
2020	917 (54%)	781 (46%)	891 (52%)	807 (48%)
2021	1019 (56%)	801 (44%)	813 (45%)	1007 (55%)
2022	1298 (59%)	902 (41%)	734 (33%)	1466 (67%)

**Table 2 healthcare-12-02031-t002:** Trends and percentage changes in hospital admissions for continuous stays and one-day stays from 2015 to 2022.

Year	Total Admissions	Change in Total Admissions (%)	Continuous Stay Admissions	Change in Continuous Stay Admissions (%)	One-Day Admissions	Change in One-Day Admissions (%)
2015	2500–2800	-	Not Provided	-	Not Provided	-
2016	2500–2800	-	Not Provided	-	Not Provided	-
2017	2500–2800	-	Not Provided	-	Not Provided	-
2018	2500–2800	-	Not Provided	-	Not Provided	-
2019	2500–2800	-	Not Provided	-	Not Provided	-
2020	1592	−36.5%	Decreased by 33.9%	−33.9%	Decreased by 76.7%	−76.7%
2021	1744	+9.5%	Gradual Recovery	Gradual Recovery	Nearly Zero	Near Zero
2022	~2500–2800	Returned to normal levels	Returned to Normal Levels	Gradual Recovery	Gradual Recovery	Gradual Recovery

**Table 3 healthcare-12-02031-t003:** Hospital admissions and gender distribution in orthopedics and traumatology cases in Bihor County.

Year	Total Admissions		Urgent Admissions		Chronic Admissions		Change Total Admissions (%)		Change Urgent Admissions (%)		Change Chronic Admissions (%)	
	M	F	M	F	M	F	M	F	M	F	M	F
2017	1479	1071	1172	1172	307	223	−4.60	12.70	−5.00	−5.00	−2.80	15.50
2018	1017	1463	800	800	217	313	−31.20	36.60	−31.70	−31.70	−29.30	40.40
2019	1378	1222	1102	1102	276	244	35.50	−16.50	37.70	37.70	27.20	−22.00
2020	917	781	757	757	155	132	−33.50	−36.10	−31.30	−31.30	−43.80	−45.90
2021	1019	801	851	851	168	132	11.10	2.60	12.40	12.40	8.40	0.00
2022	1298	902	1003	1003	295	205	27.40	12.60	17.90	17.90	75.60	55.30

M = male, F = female.

**Table 4 healthcare-12-02031-t004:** Distribution of the average length of hospitalization reported annually in correlation with COVID-19 cases in Bihor County.

Year	ALS in Orthopedics and Traumatology Departments of CCEHB (Days)	ALS in Orthopedics and Traumatology Nationwide (Days)	COVID-19 Cases in BH
2015	4.75	7.06	-
2016	4.93	7.12	-
2017	4.81	6.86	-
2018	4.80	6.78	-
2019	4.63	6.66	-
2020	3.96	6.69	17,689
2021	3.68	6.18	33,374
2022	4.10	6.08	41,099

**Table 5 healthcare-12-02031-t005:** Distribution of beds available in the orthopedics and traumatology departments reported annually in correlation with COVID-19 cases in Bihor County.

Year	Orthopedics and Traumatology Departments	COVID-19 Cases in BH	Gender	*p*
2015	63	-	-	-
2016	63	-	-	-
2017	63	-	-	-
2018	63	-	-	-
2019	63	-	-	-
2020	55	17,689	8137♀	0.002
			9552♂	
2021	63	33,374	14,685♀	0.001
			18,689♂	
2022	63	41,099	16,851♀	0.001
			24,248♂	

*p* = statistical significance.

## Data Availability

No new data were created or analyzed in this study.

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
