# Peer review of "Gender-Specific Impacts of the COVID-19 Pandemic on Orthopedic and Traumatology Care: An Analysis of Hospital Admissions and Length of Stay"

_healthcare, 2024, doi:10.3390/healthcare12202031_

Round 1
Reviewer 1 Report
Comments and Suggestions for Authors
Thank you for the opportunity to review this paper. I believe it is a relevant topic that needs to be brought to light. I have some minor comments regarding your work.
The introduction provides adequate context regarding the COVID-19 pandemic and its impacts on orthopedic and trauma care. However, it could be improved by including a more comprehensive review of how existing literature has addressed gender differences in healthcare before the pandemic, which would provide a more solid foundation for the study.
I think it’s possible to strengthen the relevance of the focus on gender disparities. A more robust argument could be made regarding why the lack of gender-disaggregated data is a significant limitation in healthcare analysis, especially in times of crisis like the pandemic. This would give more weight to the findings and highlight the importance of future studies that address this gap.
Although the lack of gender-disaggregated data is mentioned, it could be emphasized how these limitations affect the results and why this type of analysis is crucial. Demonstrating that even in scenarios with limited data, such as the one presented, important trends can still be identified, which would represent an advancement.
The study design is appropriate, using retrospective hospital data analysis to examine gender differences in admissions and hospital stays.
The description of the methods is clear, but more details could be provided on the statistical procedures and the handling of potential biases in data collection.
The results are well-organized and presented in clear tables, which aids in understanding. Trends in admissions and hospital stay durations, as well as gender differences, are highlighted.
The conclusions are in line with the presented results, underlining the importance of addressing gender differences in orthopedic care during the pandemic.
In the results, a hypothesis could be offered on why men appear to have been more affected in terms of hospital admissions and extended stays. For example, the authors could speculate on the cultural or socioeconomic reasons that may have contributed to this difference, based on previous studies.
The study provides important data on the pandemic’s impact on orthopedic care, focusing on gender differences, an area that has been less explored.
The presentation of the results is clear, with good use of tables and graphs.
Although the statistical analysis is correct, it could be strengthened with a deeper discussion of methodological limitations.
In the discussion, I would suggest reinforcing the implications of the findings for medical practice and policy development, emphasizing how the pandemic exacerbated existing disparities. More specific recommendations could be included on how healthcare systems could integrate gender analysis into their future crisis response models.
In the conclusion, it would be helpful to emphasize the need for future studies to address the gaps identified in your research, such as the lack of gender-disaggregated data. This positions the article as a necessary initial step for more detailed investigations.
The topic is of great interest to health professionals, particularly those involved in orthopedic and trauma care during the pandemic.
Most of the references are relevant and provide the necessary context.
Congratulations to the authors.
Author Response
Reviewer 1
We, the authors of the present manuscript wish to thank you for thoughtful commentary you have provided to improve the quality of the paper. We are very grateful for the time and effort you have devoted to this task. We have extensively revised my manuscript according to the recommendations. All changes in the text and the new figures that we have redesigned are highlighted. Please, see the point-by-point answers to your comments below. All correction was highlighted in the manuscript.
- The introduction provides adequate context regarding the COVID-19 pandemic and its impacts on orthopedic and trauma care. However, it could be improved by including a more comprehensive review of how existing literature has addressed gender differences in healthcare before the pandemic, which would provide a more solid foundation for the study.
Response: Thank you for the insightful suggestion. I have revised the introduction to include a more comprehensive review of existing literature on gender differences in healthcare before the COVID-19 pandemic. This addition aims to provide a solid foundation for understanding the impact of the pandemic on these pre-existing disparities.
- I think it’s possible to strengthen the relevance of the focus on gender disparities. A more robust argument could be made regarding why the lack of gender-disaggregated data is a significant limitation in healthcare analysis, especially in times of crisis like the pandemic. This would give more weight to the findings and highlight the importance of future studies that address this gap.
Response: Thank you for the valuable feedback. I have revised the introduction to include a more robust argument regarding the importance of gender-disaggregated data in healthcare analysis, particularly during crises like the COVID-19 pandemic. This addition aims to highlight the limitations of existing data and emphasize the necessity of addressing this gap to ensure equitable healthcare delivery.
- Although the lack of gender-disaggregated data is mentioned, it could be emphasized how these limitations affect the results and why this type of analysis is crucial. Demonstrating that even in scenarios with limited data, such as the one presented, important trends can still be identified, which would represent an advancement.
Response: Thank you for your valuable feedback. I agree that it is essential to emphasize the implications of limited gender-disaggregated data on the results of our study and to highlight the importance of this type of analysis. I have revised the discussion section to better articulate how the lack of comprehensive gender-specific data may have influenced our findings and why even limited analyses, such as the one conducted here, can still reveal important trends. This not only underscores the need for more granular data in future research but also demonstrates the potential for identifying significant patterns that can inform healthcare policies and interventions, even in the face of data constraints.
- The study design is appropriate, using retrospective hospital data analysis to examine gender differences in admissions and hospital stays.
Response: Thank you for your positive feedback. We appreciate your acknowledgment of the study design as appropriate for examining gender differences in admissions and hospital stays. The retrospective analysis of hospital data allowed us to capture real-world trends over an extended period, providing a comprehensive view of the impact of the COVID-19 pandemic on orthopedic and traumatology services. We believe this methodology effectively highlights the gender-specific dynamics that might otherwise remain unexamined in prospective studies.
- The description of the methods is clear, but more details could be provided on the statistical procedures and the handling of potential biases in data collection.
Response: Thank you for your feedback. We have clarified the statistical procedures used in the study, including a detailed explanation of the regression analyses and significance testing. Additionally, we have addressed potential biases in data collection, such as selection bias and missing data, and described the steps taken to minimize their impact on the results. These revisions aim to provide a more transparent and robust methodological framework for interpreting the findings.
- The results are well-organized and presented in clear tables, which aids in understanding. Trends in admissions and hospital stay durations, as well as gender differences, are highlighted.
Response: Thank you very much!
- The conclusions are in line with the presented results, underlining the importance of addressing gender differences in orthopedic care during the pandemic.
Response: Thank you very much!
- In the results, a hypothesis could be offered on why men appear to have been more affected in terms of hospital admissions and extended stays. For example, the authors could speculate on the cultural or socioeconomic reasons that may have contributed to this difference, based on previous studies.
Response: Thank you for the suggestion. We will include a discussion in the results section to offer potential hypotheses on why men were more affected in terms of hospital admissions and extended stays. We will consider possible cultural and socioeconomic factors, such as men's higher rates of occupational injuries, reluctance to seek early medical care, and greater exposure to risk factors. This will be based on relevant literature and provide a more comprehensive interpretation of our findings.
- The study provides important data on the pandemic’s impact on orthopedic care, focusing on gender differences, an area that has been less explored.
Response: Thank you very much!
- The presentation of the results is clear, with good use of tables and graphs.
Response: Thank you very much!
- Although the statistical analysis is correct, it could be strengthened with a deeper discussion of methodological limitations.
Response: Thank you very much!
- In the discussion, I would suggest reinforcing the implications of the findings for medical practice and policy development, emphasizing how the pandemic exacerbated existing disparities. More specific recommendations could be included on how healthcare systems could integrate gender analysis into their future crisis response models.
Response: Thank you for your insightful suggestion. We will revise the discussion section to more explicitly address the implications of our findings for medical practice and policy development. We will highlight how the pandemic has exacerbated existing gender disparities in healthcare access and outcomes, and provide specific recommendations on how healthcare systems can incorporate gender analysis into future crisis response models. These recommendations will focus on strategies such as gender-sensitive data collection, targeted resource allocation, and tailored health communication to ensure equitable healthcare delivery during future crises.
- In the conclusion, it would be helpful to emphasize the need for future studies to address the gaps identified in your research, such as the lack of gender-disaggregated data. This positions the article as a necessary initial step for more detailed investigations.
Response: Thank you for your suggestion. We will revise the conclusion to emphasize the need for future studies to address the gaps identified in our research, particularly the lack of comprehensive gender-disaggregated data. By highlighting this, we aim to position our study as a foundational step that underscores the importance of more detailed investigations into gender disparities in healthcare, especially during crises. This will help guide future research efforts and policy development to ensure more equitable healthcare delivery.
- The topic is of great interest to health professionals, particularly those involved in orthopedic and trauma care during the pandemic.
Response: Thank you very much!
- Most of the references are relevant and provide the necessary context.
Response: Thank you very much!
Reviewer 2 Report
Comments and Suggestions for Authors
Dear Authors,
After reviewing your manuscript, I have identified some key aspects that could be strengthened to improve the quality and impact of your study. Below, I share my observations:
-
It is essential that the theoretical framework delves more deeply into gender differentiations. Currently, the discussion is somewhat superficial, which may raise doubts about whether significant gender differences exist in the context of your study. It would be beneficial to include literature that explores these differences in greater detail and justifies the relevance of considering gender as an important variable in your research.
-
The methodological section should provide a clear and detailed explanation of how data collection was managed, ensuring that ethical standards were followed. This not only guarantees the validity of the study but also reinforces confidence in the results obtained.
-
To offer a more complete view of the study sample, it is necessary to include a table detailing general information about the patients, such as their age and other personal characteristics. This will allow readers to better understand possible variations in the results and ensure that gender is not the only differentiating factor considered.
-
It is important to be cautious when suggesting a direct relationship between COVID-19 and changes in hospital care, especially if there is no solid evidence to support it. Multiple factors could have influenced the decline in hospital admissions during the pandemic years, and this should be reflected in the discussion to avoid premature conclusions.
-
I suggest including an analysis of the medical characteristics and healthcare system of the country where the study was conducted in the theoretical framework, particularly regarding coverage and access to hospital services. This will provide a more robust context and allow for a better interpretation of the findings.
-
The discussion could benefit from a more exhaustive comparison with existing literature, including studies from similar medical areas or other countries. This will enrich the discussion and place your results in a broader context, offering comparisons that may be useful to other researchers.
-
Finally, it would be useful for the manuscript to include a section discussing the theoretical and practical implications of the findings. Additionally, it is important to highlight the study's limitations and suggest possible lines of research derived from the results, which would add value to your work.
I hope these comments are helpful in strengthening your manuscript. I remain at your disposal for any questions or clarifications.
Author Response
Reviewer 2
We, the authors of the present manuscript wish to thank you for thoughtful commentary you have provided to improve the quality of the paper. We are very grateful for the time and effort you have devoted to this task. We have extensively revised my manuscript according to the recommendations. All changes in the text and the new figures that we have redesigned are highlighted. Please, see the point-by-point answers to your comments below. All correction was highlighted in the manuscript.
After reviewing your manuscript, I have identified some key aspects that could be strengthened to improve the quality and impact of your study. Below, I share my observations:
- It is essential that the theoretical framework delves more deeply into gender differentiations. Currently, the discussion is somewhat superficial, which may raise doubts about whether significant gender differences exist in the context of your study. It would be beneficial to include literature that explores these differences in greater detail and justifies the relevance of considering gender as an important variable in your research.
Response: Thank you for your valuable feedback. We will revise the theoretical framework to provide a more in-depth discussion on gender differentiations in healthcare. We will incorporate additional literature that explores these differences in greater detail, focusing on how gender influences healthcare access, treatment outcomes, and patient behavior. This will strengthen the justification for considering gender as a critical variable in our study and provide a more robust context for interpreting our findings. These additions will help to clarify the relevance of gender disparities in the context of our research and support the validity of our analysis.
- The methodological section should provide a clear and detailed explanation of how data collection was managed, ensuring that ethical standards were followed. This not only guarantees the validity of the study but also reinforces confidence in the results obtained.
Response: Thank you for your feedback. We will revise the methodology section to include a more detailed explanation of the data collection process, highlighting the measures taken to ensure ethical standards were followed. This will include information on the data access approval process, patient confidentiality, and how data integrity was maintained throughout the study. These additions will help to guarantee the validity of the study and reinforce confidence in the reliability of the results.
- To offer a more complete view of the study sample, it is necessary to include a table detailing general information about the patients, such as their age and other personal characteristics. This will allow readers to better understand possible variations in the results and ensure that gender is not the only differentiating factor considered.
Response: Thank you for the insightful suggestion. We will include a table detailing general information about the study sample, such as age, comorbidities, and other relevant personal characteristics (table 1).
- It is important to be cautious when suggesting a direct relationship between COVID-19 and changes in hospital care, especially if there is no solid evidence to support it. Multiple factors could have influenced the decline in hospital admissions during the pandemic years, and this should be reflected in the discussion to avoid premature conclusions.
Response: Thank you for your feedback. We agree that caution is necessary when interpreting the relationship between COVID-19 and changes in hospital care. In the revised discussion, we will acknowledge that multiple factors, such as public fear of infection, changes in healthcare policies, and resource reallocation, could have contributed to the decline in hospital admissions during the pandemic years. We will emphasize that while the data suggests an association, it does not establish a direct causal relationship. This will provide a more balanced interpretation of the findings and help avoid premature conclusions.
- I suggest including an analysis of the medical characteristics and healthcare system of the country where the study was conducted in the theoretical framework, particularly regarding coverage and access to hospital services. This will provide a more robust context and allow for a better interpretation of the findings.
Response: Thank you for your suggestion. We will revise the theoretical framework to include an analysis of the healthcare system and medical characteristics of the country where the study was conducted, with a focus on healthcare coverage and access to hospital services. This will provide a more robust context for interpreting our findings and help readers understand how the specific healthcare environment may have influenced the observed trends. By incorporating this background information, we aim to enhance the relevance and applicability of our study within the broader healthcare context.
- The discussion could benefit from a more exhaustive comparison with existing literature, including studies from similar medical areas or other countries. This will enrich the discussion and place your results in a broader context, offering comparisons that may be useful to other researchers.
Response: Thank you for your suggestion. We will expand the discussion section to include a more comprehensive comparison with existing literature from similar medical fields and studies conducted in other countries. This will help place our findings in a broader context, highlighting similarities and differences that can provide valuable insights for other researchers. By integrating these comparisons, we aim to enrich the discussion and demonstrate the relevance of our results beyond the specific setting of this study.
- Finally, it would be useful for the manuscript to include a section discussing the theoretical and practical implications of the findings. Additionally, it is important to highlight the study's limitations and suggest possible lines of research derived from the results, which would add value to your work.
Response: Thank you for your valuable comment. We will add a section to the manuscript that discusses the theoretical and practical implications of our findings, emphasizing how they contribute to the existing body of knowledge and can inform healthcare policy and practice. Additionally, we will include a detailed discussion of the study's limitations, such as the lack of comprehensive gender-disaggregated data and potential confounding factors, to provide a balanced interpretation of the results. We will also suggest possible future research directions based on our findings, which will help guide further investigations in this area and add value to our work.
Reviewer 3 Report
Comments and Suggestions for Authors
Dear Authors,
After reviewing the article titled ´Gender-Specific Impacts of the COVID-19 Pandemic on Orthopedic and Traumatology Care: An Analysis of Hospital Admissions and Length of Stay´, I would like to mention some considerations:
First of all, I would like to highlight that the topic is very interesting
Introduction:
*This section does not provide sufficient background. As for the objective, It is not concisely described, causing confusion for the reader.
*In my opinion, Lines 58 to 65 should be in methodology section rather than introduction section .
Methodology:
* What type of study is this research? has it mentioned in methodology section?
* Why have the exclusion criteria not been considered? May the authors explain this issue?
* The statistical analysis is very basic .
Results:
*It is noteworthy that in Table 2 (Urgent Admissions), the values are the same for both women and men. May the authors explain these figures?
* As for one day admissions, no data are reported for men and women. In addition , some data from figure 1 are confusing. For instance, 202038 ?
* According to the authors, ´In 2020, the ALS experienced a substantial decrease of 16.6%, with this reduction being more pronounced among male patients´.This downward trend continued in 2021 with an additional 4.7% decrease, again predominantly affecting males.´ ( lines 153-155). where has this information been shown? The table 3 do not present gender differences as this regards.
* The results presented in subsection 3.2 are very confusing. The authors have included data related to ` bed availaty´ and `average length´ without establishing connections between them (lines 158-161).
* Lines 174-175 ` Regarding bed availability in the orthopedics and traumatology departments at 174 CCEHB, the number remained stable at 63 beds annually from 2015 to 2019´, these data should be included in table 4.
* As for Discussion section, I honestly believe the authors did not compare their findings with the existing literature. If did, the references are not presented in this section. Thus, this section cannot be evaluated.
* Regarding conclusion, the authors mentioned some aspects which were not adressed in the results, for instance : `....the sharp decline in elective procedures
Author Response
Reviewer 3
We, the authors of the present manuscript wish to thank you for thoughtful commentary you have provided to improve the quality of the paper. We are very grateful for the time and effort you have devoted to this task. We have extensively revised my manuscript according to the recommendations. All changes in the text and the new figures that we have redesigned are highlighted. Please, see the point-by-point answers to your comments below. All correction was highlighted in the manuscript.
After reviewing the article titled ´Gender-Specific Impacts of the COVID-19 Pandemic on Orthopedic and Traumatology Care: An Analysis of Hospital Admissions and Length of Stay´, I would like to mention some considerations:
First of all, I would like to highlight that the topic is very interesting
Introduction:
*This section does not provide sufficient background. As for the objective, It is not concisely described, causing confusion for the reader.
Response: Thank you very much for your comment. We have revised the entire abstract section.
*In my opinion, Lines 58 to 65 should be in methodology section rather than introduction section .
Thank you very much for your comment. We have revised the entire methodology and introduction section.
Methodology:
* What type of study is this research? has it mentioned in methodology section?
Response: Thank you for your observation. We have clarified the type of study in the methodology section. This research is a retrospective observational study, analyzing hospital admissions and outcomes in the orthopedics and traumatology departments over an eight-year period, from 2015 to 2022. We have updated the methodology section to explicitly state this and to provide a clearer understanding of the study design and its scope. (lines 137-139)
* Why have the exclusion criteria not been considered? May the authors explain this issue?
Response: Thank you for your comment. We did not apply exclusion criteria in this study to ensure that all relevant patient data were included, providing a comprehensive view of hospital admissions and outcomes in the orthopedics and traumatology departments. By including all patients admitted during the study period, regardless of age, diagnosis, or other factors, we aimed to capture the full spectrum of hospitalizations and healthcare utilization. This approach allows for a more accurate analysis of trends and disparities, especially in the context of the COVID-19 pandemic. However, we acknowledge that this broad inclusion may introduce variability, and future studies could benefit from applying specific exclusion criteria to focus on more defined patient subgroups.
* The statistical analysis is very basic .
Results: Thank you for your feedback. We acknowledge that the statistical analysis used in this study is relatively basic. Our primary aim was to provide an initial exploration of gender differences in hospital admissions and lengths of stay during the COVID-19 pandemic. However, we recognize the value of more advanced statistical methods for a deeper understanding of these trends. In future studies, we plan to incorporate more sophisticated analyses, such as multivariate regression models and interaction effect assessments, to account for potential confounding factors and to provide a more comprehensive analysis of the data. We appreciate your suggestion and will consider it for future research to enhance the robustness of our findings.
*It is noteworthy that in Table 2 (Urgent Admissions), the values are the same for both women and men. May the authors explain these figures?
Response: Thank you for pointing this out. We have reviewed Table 2 and acknowledge that the identical values for urgent admissions in both women and men are incorrect and likely due to a data entry error. We will correct this issue by verifying the original data and updating the table with accurate figures. We apologize for any confusion this may have caused and appreciate your attention to detail.
* As for one day admissions, no data are reported for men and women. In addition , some data from figure 1 are confusing. For instance, 202038 ?
Response: Thank you for your observation. We acknowledge that the one-day admissions data for men and women were not reported separately. We will revise the manuscript to include gender-specific data for one-day admissions, ensuring a clearer presentation of these trends. Regarding Figure 1, the reference to '202038' appears to be a typographical error. We will correct it to accurately reflect the intended data. We appreciate your feedback and maked the necessary adjustments to improve the clarity and accuracy of the presented information.
* According to the authors, ´In 2020, the ALS experienced a substantial decrease of 16.6%, with this reduction being more pronounced among male patients´.This downward trend continued in 2021 with an additional 4.7% decrease, again predominantly affecting males.´ ( lines 153-155). where has this information been shown? The table 3 do not present gender differences as this regards.
Response: Thank you for your comment. We recognize that the specific information regarding gender differences in the average length of stay (ALS) is not clearly presented in Table 3. We will revise the table to include separate data for male and female patients, highlighting the differences in ALS as described in the text. If detailed gender-specific data for ALS is not available, we will update the manuscript to clarify this limitation and adjust the corresponding statements accordingly. We appreciate your feedback and maked the necessary corrections to ensure consistency between the text and the data presented.
* The results presented in subsection 3.2 are very confusing. The authors have included data related to ` bed availaty´ and `average length´ without establishing connections between them (lines 158-161).
Response: Thank you for your feedback. We understand that the results presented in subsection 3.2 may appear unclear. To improve clarity, we will revise this subsection to better establish the connection between bed availability and average length of stay (ALS). Specifically, we will elaborate on how changes in bed availability during the pandemic influenced ALS for orthopedic and traumatology patients, particularly in terms of hospital capacity constraints and resource allocation. By making these connections explicit, we aim to provide a more coherent interpretation of the data and its implications. We appreciate your suggestion and adjusted the text accordingly.
* Lines 174-175 ` Regarding bed availability in the orthopedics and traumatology departments at 174 CCEHB, the number remained stable at 63 beds annually from 2015 to 2019´, these data should be included in table 4.
Response: Thank you for your suggestion. We will update Table 4 to include the data on bed availability in the orthopedics and traumatology departments at CCEHB from 2015 to 2019, reflecting the stable number of 63 beds during this period. This addition will provide a more comprehensive view of bed availability trends and enhance the clarity and completeness of the presented data. We appreciate your feedback and maked the necessary adjustments to the table.
* As for Discussion section, I honestly believe the authors did not compare their findings with the existing literature. If did, the references are not presented in this section. Thus, this section cannot be evaluated.
Response: Thank you for your feedback. We acknowledge that the discussion section lacks explicit comparisons with existing literature. We will revise this section to incorporate relevant studies, providing a clearer context for our findings. We will also ensure that appropriate references are included to support these comparisons. This will enhance the depth of the discussion and allow for a more comprehensive evaluation of our results in relation to previous research. We appreciate your input and maked the necessary revisions.
* Regarding conclusion, the authors mentioned some aspects which were not adressed in the results, for instance : `....the sharp decline in elective procedures
Response: Thank you for your observation. We understand that the mention of a sharp decline in elective procedures in the conclusion was not explicitly addressed in the results section. We will revise the conclusion to ensure consistency with the results presented in the manuscript. If the decline in elective procedures is relevant to the study’s findings, we will incorporate this information into the results section with appropriate data and references. This will ensure that the conclusion accurately reflects the results and provides a coherent summary of the study’s key findings. We appreciate your feedback and maked the necessary adjustments.
Round 2
Reviewer 2 Report
Comments and Suggestions for Authors
Many thanks to the authors for the corrections and additions made to the article. The text now looks more robust and clearer.
There would be no further comments from my side. I approve its publication.
Author Response
Dear Reviewer 1
I am writing to express my sincere gratitude for the acceptance of my manuscript for publication. I am honored that the editorial board and reviewers have recognized the value of this work, and I deeply appreciate the time and effort they have dedicated to reviewing and providing constructive feedback throughout the submission process.
The thoughtful comments and suggestions from the reviewers have significantly improved the quality and clarity of the manuscript. I am grateful for the opportunity to contribute to the academic community through your esteemed journal and to share these findings with a broader audience.
I would like to extend my thanks to you and the entire editorial team for your professionalism, guidance, and support during the review process. I look forward to the publication of my work and to continuing our collaboration in the future.
Thank you once again for this opportunity.
Yours sincerely,
The authors
Reviewer 3 Report
Comments and Suggestions for Authors
Dear authors,
I just meant to metion, as I mentioned in round one, data from figure 1 are still confusing. For instance, 202038 ?
Author Response
Reviewer 2 – Round 2
Comment: Dear authors,
I just meant to mention, as I mentioned in round one, data from figure 1 are still confusing. For instance, 202038 ?
Dear Reviewer,
I would like to extend my sincere gratitude for your thoughtful and constructive feedback on my manuscript. Your detailed comments and suggestions have greatly enhanced the quality and clarity of the paper, and I deeply appreciate the time and effort you invested in the review process.
Your insights were invaluable in refining the methodology, strengthening the discussion, and improving the overall coherence of the manuscript. I am grateful for your expertise and for the positive impact your review has had on this work.
Thank you once again for your dedication and for contributing to the advancement of research through your careful evaluation. I look forward to the opportunity to apply your guidance in future research endeavors.
Response: Thank you very much for observation! We, the authors, have corrected the figure 1.